# DiffImpact: Differentiable Rendering and Identification of Impact Sounds

**Samuel Clarke**    **Negin Heravi**    **Mark Rau**    **Ruohan Gao**
**Jiajun Wu**    **Doug James**    **Jeannette Bohg**
Stanford University
{spclarke, nheravi, mrau, rhgao, jiajunw, djames, bohg}@stanford.edu

**Abstract:** Rigid objects make distinctive sounds during manipulation. These sounds are a function of object features, such as shape and material, and of contact forces during manipulation. Being able to infer from sound an object's acoustic properties, how it is being manipulated, and what events it is participating in could augment and complement what robots can perceive from vision, especially in case of occlusion, low visual resolution, poor lighting, or blurred focus. Annotations on sound data are rare. Therefore, existing inference systems mostly include a sound renderer in the loop, and use analysis-by-synthesis to optimize for object acoustic properties. Optimizing parameters with respect to a non-differentiable renderer is slow and hard to scale to complex scenes. We present DiffImpact, a fully differentiable model for sounds rigid objects make during impacts, based on physical principles of impact forces, rigid object vibration, and other acoustic effects. Its differentiability enables gradient-based, efficient joint inference of acoustic properties of the objects and characteristics and timings of each individual impact. DiffImpact can also be plugged in as the decoder of an autoencoder, and trained end-to-end on real audio data, so that the encoder can learn to solve the inverse problem in a self-supervised way. Experiments demonstrate that our model's physics-based inductive biases make it more resource efficient and expressive than state-of-the-art pure learning-based alternatives, on both forward rendering of impact sounds and inverse tasks such as acoustic property inference and blind source separation of impact sounds. Code and videos are at https://sites.google.com/view/diffimpact.

**Keywords:** Differentiable Sound Rendering, Auditory Scene Analysis

## 1  Introduction

The sound we perceive during rigid object contact is a function of many factors, including noise in the environment, position of the listener, reflecting surfaces in the surrounding environment, etc. The most defining of these factors are the vibrations the object makes, a product of its intrinsic acoustic properties and the way it was contacted. Extracting information about the acoustic properties of an object could inform a robot about the object's size, shape, material, and where and how it was struck. Extracting information about the way the object was contacted could inform a robot of the velocity, mass, shape, and hardness of the tool that was used to strike the object. Therefore, the ability to interpret impact sounds could augment and complement what robots can perceive from vision, especially in cases of occlusion, low visual resolution, poor lighting, or blurred focus.

Decomposing the sound an object makes into the contact forces that excited it and the idealized acoustic impulse response is generally an ill-posed problem. As annotations on sound data are rare, existing inference systems mostly include a sound renderer in the loop, and use analysis-by-synthesis to optimize for object acoustic properties. Optimizing parameters with respect to a non-differentiable renderer is slow and hard to scale to complex scenes. We propose DiffImpact, a fully differentiable, generative sound model that uses physics-based priors as a bias. Specifically, we introduce fully differentiable models of the object's impulse response and the contact force during impact. We combine these with differentiable models of environment acoustics such as ambient noise and reverberation into a fully differentiable generative sound model. With this physics-based model, we can now infer physically interpretable parameters of rigid body impact sounds through analysis-by-synthesis, even

5th Conference on Robot Learning (CoRL 2021), London, UK.

from real-world recordings, including from real YouTube videos or from a noisy and reverberant robotics lab.

Impacts in the real world are not always clean impulses, nor do they occur in controlled acoustic environments. We show how DiffImpact's differentiability allows us to learn meaningful sound models that are fit to the sounds of multiple imperfect impact sounds over one or multiple recordings. We first validate our approach on single impact sounds. We show that our model can fit interpretable physical parameters of an object's sound model and of the contact forces to audio recordings. We then show how DiffImpact enables a robot to passively learn object sound models from a real-world dataset (ASMR YouTube videos [1]). Our model not only extracts physically interpretable parameters, but also outperforms model-based and model-free learning baselines in rendering realistic impact sounds for held-out data. Finally, we show how a robot can learn object sound models from data it collected by striking these objects with a tool. Specifically, we show how the robot can perform blind source separation of the collected sound and thereby achieves 140% higher accuracy on classifying the material of the objects it strikes.

## 2 Related Work

Audio-frequency vibrations have been shown to be an effective modality for robots to ascertain characteristics of physical events in their environment, including instance level classification of everyday objects [2, 3] and estimates of their motion [4], terrain classification by legged robots [5], masses of granular materials being poured from a scoop the robot is holding [6], and amounts of liquid poured by a robot [7]. Auditory scene analysis also has applications for dynamic inference of stochastic events, enabling creation of audio based reactive robotic trackers [8].

Many of these works have a common characteristic with more recent visual perception approaches, that learning-based neural methods are subsuming many classical approaches by enabling automatic feature extraction rather than requiring hand engineered features [9, 10]. However, whereas differentiable image renderers have empowered new approaches in visual perception [11, 12, 13, 14], differentiable audio renderers have been introduced only recently. Engel et al. [15] recently debuted the Differential Digital Signal Processing (DDSP) library, showing applications in the musical domain on tasks such as disentangling the pitch and timbre of instrument sounds. Rather than the musical domain, we focus on modeling modal impact sounds of rigid objects, and show applications of our physics-based model for robotics tasks.

Model-free neural audio representation methods are powerful in synthesizing realistic human speech [16, 17] but lack interpretability and flexibility for transfer to different categories such as transient environment sounds. Model-based methods often require less data and training time to produce interpretable learned parameters, which can be reused and applied to other tasks more intuitively. They are also capable of synthesizing realistic object sounds [18, 19, 20] but depend on hand-engineered analytical methods for parameter estimation with structured audio samples. Ren et al. [21] use real audio clips of striking objects to infer properties such as Young's modulus and the modes of the object, then attempt to reproduce the sound based on a modal model with a residual. However, they assume knowledge of the 3D geometry of the object as well as impact location. Zhang et al. [22] infer object properties such as Young's modulus and Rayleigh damping from sounds of objects falling on a surface by using a physics-based modal sound generation engine with a black box forward model. But their model is not fully differentiable, making it challenging to learn from unconstrained in-the-wild audio or to optimize over many dimensions at once. While we use a model conceptually similar to that of [23, 22], our fully differentiable structure empowers learning a best estimate of an object's modal model from the sound of multiple uncontrolled, imperfect, heterogeneous impacts, rather than requiring strictly controlled recordings for analysis.

## 3 Differentiable Rendering of Impact Sounds

Our DiffImpact models the sound that a rigid object makes when struck in a real environment as a function of 1) the impact force's magnitude over time; 2) the brief click of a sound pressure wave caused by the rapid deceleration of the striking object, referred to as the "acceleration sound"; 3) the struck and striking objects' impulse responses, or their idealized theoretical vibrational response to a perfect unit impulse; 4) any environment, background, or recording noise; and finally 5) the impulse response of the local environment, reflecting and transmitting reverberations of any sound being directly emitted by the object. DiffImpact attempts to decompose real impact sounds into each of these contributions, using physics-based models in order to regularize this decomposition.

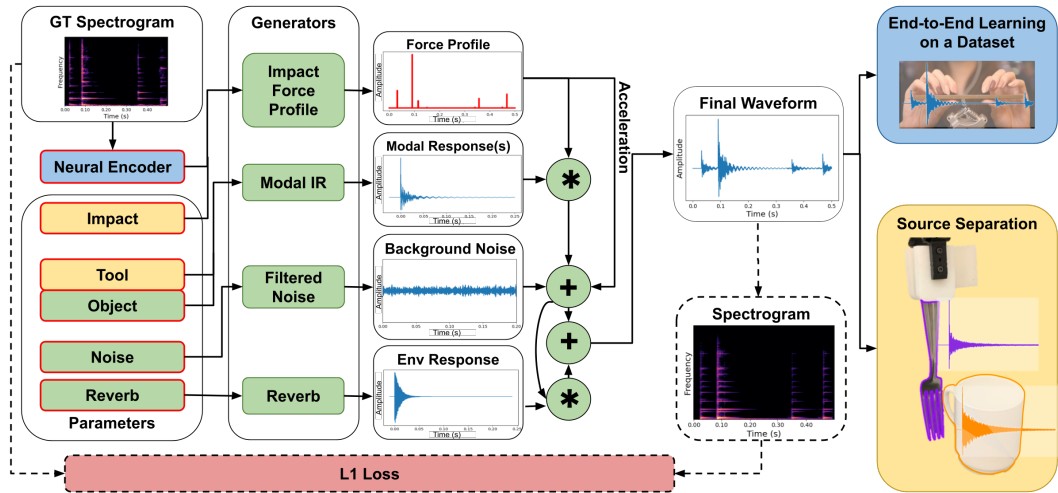

**Figure 1:** Overview of proposed approach, DiffImpact. Blue elements are only used for the end-to-end learning task, yellow elements are only used in source separation, and green elements are used in both. Dashed components are only used during training, and elements with a red outline have learnable parameters.

## 3.1 Overview

Figure 1 provides a high-level overview of our approach. There are some model variations dependent on which inference task we are considering. For clarity, here we will give a brief overview of the model when used for the end-to-end learning task of predicting an audio waveform from a magnitude spectrogram of the ground truth audio sample, similar to the models in [15, 24, 16]. The magnitude spectrogram lacks phase information and is therefore not directly invertible to an audio waveform. This spectrogram is input to an encoder network that outputs a coarsely temporally discretized time series of parameters that represent impact characteristics. These parameters are the input to the *Impact Force Profile Generator* that synthesizes an impact force profile as described in more detail in Section 3.2. Given object-specific, learnable parameters of a modal model of rigid body impact sound, the *Modal Impulse Response Generator* synthesizes an impulse response as described in more detail in Section 3.3. This impulse response is convolved with the force profile to synthesize object vibrations. To model the contribution of the sound pressure wave caused by the rapid deceleration of the impacting object, this impact force profile is also multiplied by a learnable scalar and added to the object vibration sounds.

We theoretically now have all the sounds emitted directly from the point of impact on the object. To model sound that is specific to the environment, we introduce a noise and reverb generator as detailed in Section 3.4. Given learnable noise parameters, the *Filtered Noise Generator* synthesizes recording or background noise that is then added to the current audio. Finally, given learnable reverb parameters, the *Reverb Generator* synthesizes an impulse response of the environment. This response is then convolved with the current audio to model the sounds that have transmitted through the environment and reflected off nearby surfaces, rather than directly from the object. This result is added to the current audio to produce the final waveform.

For training the aforementioned learnable parameters of the generators and the encoder, we use a loss that compares spectrograms of the generated waveform with those of the ground truth (Section 3.5).

The final waveform $\hat{W}(t)$ produced by our model can be expressed as

$$\hat{W}(t) = [F(t; \mathbf{m}, \mathbf{t}, \boldsymbol{\tau}) \circledast [\alpha + IR_o(t_{IR}; \mathbf{f}, \mathbf{g_o}, \mathbf{d_o})] + N(t; \mathbf{g_n})] \circledast (1 + IR_e(t_{IR}; \mathbf{g_e}, \mathbf{d_e})), \quad (1)$$

where $t$ is the time in the final waveform; $F$ is the force profile function that takes as input the impulse magnitudes $\mathbf{m}$, impulse timings $\mathbf{t}$, and contact time scales $\boldsymbol{\tau}$; $\circledast$ denotes a convolution; $\alpha$ is a scalar for the acceleration sound; $IR_o$ is the modal object impulse response function parameterized by modal frequencies $\mathbf{f}$, gains $\mathbf{g_o}$, and dampings $\mathbf{d_o}$, and $t_{IR}$ is time within each impulse response; $N$ is the filtered noise parameterized by frequency band gains $\mathbf{g_n}$; and $IR_e$ is the impulse response of the environment parameterized by frequency band gains $\mathbf{g_e}$ and dampings $\mathbf{d_e}$. We will now detail the equations and physics-based models for each of these components we propose, and how we scale a low dimensional set of normalized parameters into input parameters for each of these functions, producing an audio frequency signal in a differentiable manner.

## 3.2 Differentiable impact force profile and acceleration sound

To model the force profiles of multiple impact events occurring over the time interval of an audio clip, our proposed module converts the two time series outputs from the encoder into a contact force profile at audio frequency. This profile is defined by the time scales $\boldsymbol{\tau}$ and magnitudes $\mathbf{m}$ of the impulses.

First, we define a physical model for each impulse event. Since objects will generally have convex curvature at the points at which they strike, they can be locally approximated as spheres in contact. Therefore, we adopt a Hertz half-sine contact model based on collisions between frictionless rigid spheres similar to that of [25]. To ensure smoothness, we use a Gaussian approximation $C_i(t;t_i,\tau_i)$ of this contact model force impulse profile [26, 27]:

$$C_i(t;t_i,\tau_i) = \exp\left(-\frac{6}{\tau_i^2}\left(t-t_i-\frac{\tau_i}{2}\right)^2\right), \tag{2}$$

where $t_i$ represents the timing of the onset of the unsmoothed impact, and $\tau_i$ is the time scale of the contact force, which depends on the velocity of the impact and material properties of the object [28]. Sharper and faster contact points as well as harder materials will each contribute to reducing the value of $\tau_i$, while softer, slower, and duller contact points will have higher $\tau_i$ values.

Equation 2 provides a model for an individual contact event. But in a natural audio clip, multiple such contact events, each with different characteristics, can occur over a given interval, such that a natural force profile over time will be the sum of multiple such contact impulses, each with different parameters and magnitudes, and each occurring at different instants during the interval. We model the force profile, the scalar value of the normal force over time, for multiple contacts as a weighted sum of such Gaussian impulses from Equation 2:

$$F(t;\mathbf{m},\mathbf{t},\boldsymbol{\tau}) = \mathbf{m}^T \mathbf{C}(t;\mathbf{t},\boldsymbol{\tau}), \tag{3}$$

where $\mathbf{t} = [t_0,...,t_M]^T$, $\boldsymbol{\tau} = [\tau_0,...,\tau_M]^T$ and $\mathbf{m} = [m_0,...,m_M]^T$. Each magnitude $m_i$ serves as the weight for the corresponding impact. $\mathbf{C}(t;\mathbf{t},\boldsymbol{\tau})$ stacks the output of Equation 2 into a vector.

Equation 2 is naturally differentiable, but a challenge to making Equation 3 fully differentiable is that the number of impacts occurring during an interval may not be be known a priori, and may vary between examples. To address this, we propose an algorithm that finds $M$ peaks in a time series of impact magnitude estimates over time, and places model-based impulses at those peaks based on the parameters at each peak's point in the time series. If there are fewer than $M$ peaks, the impact magnitudes $m_i$ at the extraneous peaks should be estimated as zero. We detail this algorithm in Appendix A.1.

To model the acceleration sound, we must know the acceleration profile of the striking tool's tip. Since the striking tool's mass is fixed, then the deceleration is proportional to the impulse force, which is already estimated in Equation 3. Therefore, we model the contribution of the acceleration sound as the force profile from Equation 3 multiplied by the learnable scalar variable $\alpha$.

## 3.3 Object impulse response

For the object impulse response, we adopt a modal model by modeling objects as spring-damper systems, where the impulse response is a linear combination of exponentially decaying sinusoids, with each sinusoid having a different frequency $f_k$, gain $g_k$, and damping $d_k$ [18, 23]. We model the $K$ most salient modes of the object through constructing a finite impulse response (FIR) as $IR_o(t;\mathbf{f},\mathbf{g_o},\mathbf{d_o}) = \mathbf{g_o}^T[\exp(-\mathbf{d_o}t) \circ \sin(2\pi\mathbf{f}t)]$, where $\mathbf{f} = [f_0,...,f_K]$, $\mathbf{g_o} = [g_0,...,g_K]$, $\mathbf{d_o} = [d_0,...,d_K]$ and $\circ$ is the Hadamard product. Note that we make the simplifying assumption that the gains of each object are fixed across the entire body of the object, whereas in reality for general objects, they vary significantly depending on the location at which the object is impacted. We made this assumption to make the optimizations in our experiments more tractable since the inputs to our frameworks were blind to the 3D models of the objects as well as the locations of each contact. We detail how we scale normalized inputs to our module to produce the parameters in Appendix A.2.

## 3.4 Environment noise and reverberation

We approximate background and measurement noise as static filtered noise, whereas we approximate reverberation from the measurement environment with exponentially decaying filtered noise. For background and measurement noise, our filtered noise function $N(t;\mathbf{g_n})$ filters random white noise with time-constant gains for each frequency band given by learnable noise parameters: $N(t;\mathbf{g_n}) = \mathbf{g_n}^T filt(\mathcal{N}(t),B_n)$, where $filt(\mathcal{N}(t),B_n) \in \mathbb{R}^{B_n \times T}$ is a function that filters white noise $\mathcal{N}(t)$ with

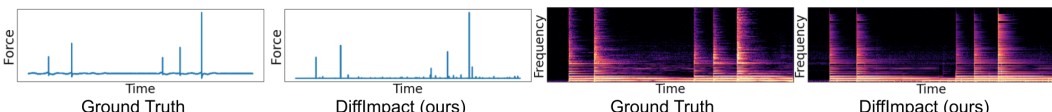

**Figure 2:** Comparing our model's estimates to ground truth time series of normalized contact forces and spectrograms, for our analysis by synthesis task on short recordings. Results are from the steel bowl.

$B_n$ band pass filters spaced linearly over the interval of frequencies to the Nyquist frequency. The output matrix is multiplied with the vector $\mathbf{g_n}$ weighting each frequency band.

For generating reverberations, we model the local environment's response to sounds with an impulse response with exponentially decaying filtered noise, where each noise band has its own decay rate, similar to [23]: $IR_e(t_{IR};\mathbf{g_e},\mathbf{d_e}) = \mathbf{g_e}^T[\exp(-\mathbf{d_e}t) \circ filt(\mathcal{N}(t), B_e)]$. Here $\mathbf{g_e}$ and $\mathbf{d_e}$ are the gains and damping factors per frequency band, respectively, and have dimension $B_e$. We describe how these parameters are derived from scaling our learnable normalized reverb parameters in Appendix A.3.

### 3.5   Loss function

As a loss function for all tasks, we use the sum of the $\mathcal{L}_1$ distance between the spectrograms and log spectrograms of multiple window sizes from the ground truth $W$ and synthesized audio $\hat{W}$ [15, 24]:

$$\mathcal{L}(W,\hat{W}) = \sum_{s_w \in \{128,\dots,2048\}} \lambda_1 |S(W,s_w,\beta s_w) - S(\hat{W},s_w,\beta s_w)| + \lambda_2 |\log(S(W,s_w,\beta s_w)) - \log(S(\hat{W},s_w,\beta s_w))|,$$

where $s_w$ is the window size, $\beta$ is the overlap ratio, $\lambda_1$ and $\lambda_2$ are weights, and $S$ is the short-term Fourier transform (STFT) or spectrogram. Using multiple spectrogram scales ensures that the loss characterizes an error signal in both high frequency and temporal resolution. The log spectrograms also ensure that higher frequency components, which tend to have lower energy magnitude than low frequency components, are also weighted more strongly. We simultaneously fit estimates of parameters for both the sound the object makes when impacted, and the impacts that produce those sounds.

## 4   Results

We first validate that DiffImpact is able to not only learn expressive sound models, but also to extract accurate physically interpretable parameters from controlled recordings of impacting different household objects. Then we show how our contributed modules can be used in two different real-world tasks, one showing how a robot could use our models to passively learn sound models from data, and another showing how our robot can interactively learn sound models and use them for useful downstream tasks.

### 4.1   Analysis by synthesis: Fitting to single real impact sounds

To test whether our DiffImpact can extract contact forces from sound, we collected 3-second clips of striking four different household objects (bottom right of Figure 4) in a recording studio with an impact hammer (PCB 086C01). The impact hammer had a force transducer in its tip, providing ground truth contact forces synchronized with the audio recorded by a microphone (PCB 378A06). We randomly initialized the parameters for our model, and fit them to each recording with respect to our loss function which only compares synthesized and ground truth spectrograms. We use Adam as the optimizer [29]. Our results are shown in Figure 2. Though our model's estimates of contact force have no sense of scale or calibration to a physical quantity (see Appendix E), we show that the force profiles our model synthesized were quite similar in shape to those of the ground truth. Note that we do not supervise directly on force profiles. Furthermore, the spectrograms demonstrate that our model is able to synthesize audio very similar to the ground truth. Audio examples are included in the Supplementary Materials, and further details of this experiment and its setup are in Appendix B.

### 4.2   End to end learning: Learning rigid body impact sound models from in-the-wild audio

We demonstrate the utility of our modules in the end to end learning task of autoencoding real world impact audio. Similar to previous work [16, 15], the input to each model is the ground truth magnitude spectrogram, and the output is the wave form of the audio. Our experiments in this section are driven by evaluating perceptual realism and similarity of our model's predictions, measured both qualitatively by human user studies and quantitatively by perceptual loss metrics.

**Dataset**   We curated a dataset of audio clips extracted from videos on the ASMR Bakery YouTube Channel [1]. We selected clips from segments in which the creator is repeatedly tapping five objects

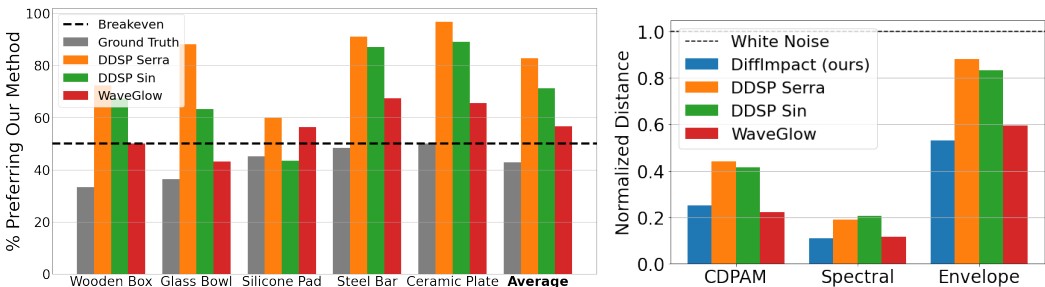

**Figure 3:** Results from end-to-end learning task evaluations on ASMR dataset. (**Left**) Human study ratings of realism of different ASMR objects' generated audio from test set. (**Right**) Average computational audio distance metrics of test set output from different models.

of different materials: a wooden box, a glass bowl, a silicone pad, a steel bar, and a ceramic plate. Screenshots from clips of each of the objects we selected are shown Appendix C.1. With this dataset, we show that a robot could use our framework to learn meaningful, physically interpretable models of rigid objects from a rich real world dataset. This dataset presents many challenges to the task of estimating a modal sound model, challenges which prior works have not addressed. First, whereas other works record the sound of a single controlled impact by a small pellet or specialized instrument [23, 18], many clips in our dataset include impacts randomly alternating between tapping with hard fingernails and soft fingertips within short timespans. Some of these objects are being held in the creator's hand while they are being tapped, and other objects are lying on a table (see Appendix C.1). Holding each object or resting it on a surface will dampen certain modes of vibration. Therefore, at no point in the entire dataset does an object emit an ideal undamped modal response. The ASMR dataset consists of around 3-5 minutes of continuously recorded audio of tapping for each object. We validated our model and its hyperparameters on a held out validation set of the steel bar clips, then split remaining data for each object in approximately a 90-10 train-test split. Thus, each model had 200 seconds of training data for each object, and at least 20 seconds of unseen audio samples per object.

**Baselines** We implemented two baseline models based on DDSP [15, 24], which we refer to as "DDSP Serra" and "DDSP Sin." DDSP Serra is based on a time-varying harmonic model developed for musical instruments [30]. To test the influence of the harmonic model bias, DDSP Sin replaces the harmonic oscillator with an unrestricted time-varying sinusoidal oscillator. We also tested WaveGlow [16], a state of the art model-free approach for generating human speech. Further details on baselines are in Appendix C.2. The WaveGlow baseline took more than two orders of magnitude more GPU time to train ($\sim$1 hour vs. $\sim$ 300 hours of training time for a single object on an Nvidia Quadro P5000 GPU). To demonstrate the necessity of this training time, we report results from training this model for only $\sim$1 order of magnitude more than the other models in Appendix C.4. For our evaluations, we trained a separate instance of each baseline and our model per object.

**Qualitative evaluation through human studies** To test DiffImpact's capability of producing realistic impact sounds on held-out audio samples of each object compared to baselines, we re-dubbed the original videos with each model's output given the video's original audio as input. We then presented participants on Amazon Mechanical Turk with a two-alternative forced choice to pick the best matching audio for 5-second videos. The comparisons were between our model and either the baselines or ground truth (1200 questions total). Our model significantly outperforms all the baselines on average (p<0.05), as shown on the left of Figure 3. Per object category, DiffImpact outperforms all baselines for the steel bar and ceramic plate. It outperforms DDSP Serra and DDSP Sin models for all objects except for the silicone pad (p<0.05). We suspect our model struggles with modeling the impact sound of the silicone pad due to the damping effect and roughness of the material as well as its high deformability that violates our model's rigid body assumption. The other models without these biases may better model its sound. Furthermore, the user's hand sometimes rubs against the surface during consequent tapping, resulting in unmodeled components in the audio which affect the performance of our method. On average, our participants chose our model's audio over ground truth 42.9% of the time, suggesting that our generated audio is often realistic enough to be confused with ground truth audio.

**Quantitative evaluation using distance metrics** We also compared each model's test outputs with the ground truth audio using different computational distance metrics, including CDPAM, a learning-based perceptual distance metric designed for human speech applications [31], the multi-scale spectrogram loss used in [15, 24], and the envelope distance used in [32]. Since each of these metrics have

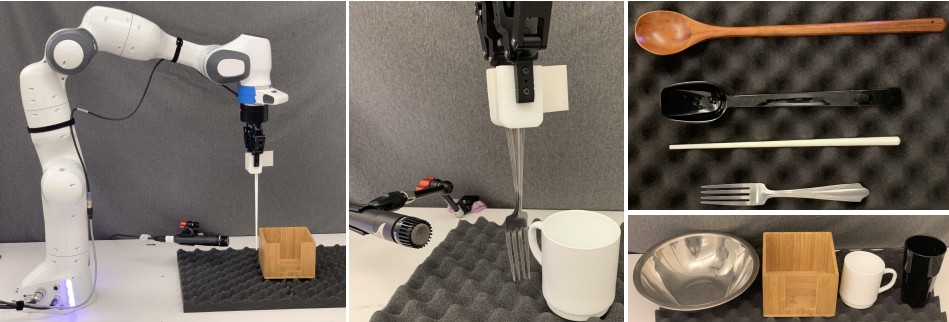

**Figure 4:** Physical setup and tools for robotic impact dataset collection. **(Left)** A Franka Panda holds and rotates a ceramic chopstick to swing at the wood napkin holder, with audio recorded by the bottom-left microphone. **(Middle)** Close up of a fork swung at a mug, with microphone at left. Objects were rested on a soft foam pad to prevent sliding and mitigate contact damping and acoustic reflection from the hard tabletop. **(Right top)** The tools used for striking (from top): wooden spoon, polycarbonate spoon, ceramic chopstick, and steel fork. **(Right bottom)** The struck objects (from left): steel bowl, wood napkin holder, ceramic mug, and polycarbonate cup.

widely different scales, we normalized each model's average distance on each metric by the average distance of white noise with respect to that metric. The results are shown on the right of Figure 3. These results align with our human studies, suggesting that our DiffImpact outperforms DDSP Serra and Sin, and it performs on par with WaveGlow, yet requires two orders of magnitude less training time.

**Discussion**   We demonstrate that DiffImpact is expressive enough to characterize and generate real world impact sounds, with performance competitive or better than state of the art alternatives. But perhaps most importantly, while completing this end-to-end learning task, our model is forced to extract physically interpretable and transferable parameters of impact forces and acoustic properties of the object and environment, whereas the parameters learned by other models are either uninterpretable (in the case of WaveGlow), or not directly meaningful to extracting impact event and intrinsic acoustic properties of objects. These parameters can be used with a novel impact profile to render the sound the object would make when impacted as such, creating a multimodal object model for simulation environments [33, 34] merely from information learned from in-the-wild YouTube videos.

### 4.3   Robotic task: Source separation of kitchen utensil sounds

We use DiffImpact to perform a source separation task. Unlike traditional audio source separation, where the goal is to isolate sounds in passive, pre-recorded audios and videos [35, 36, 37], our goal is to enable robots to actively interact with objects in the environment and separate the sounds they make. Consider the scenario where the robot strikes an object with a tool. The perceived sound of the impact will be a combination of the sounds the tool makes and the sounds the struck object makes. We show that by separating them, a robot can more accurately classify object materials, as different materials have distinct acoustic properties. Knowing the material of an object can inform the robot about its durability, surface friction, and category, all key factors to consider when planning grasps and actions.

**Dataset**   We collected a dataset of a robot swinging four household tools ("tools") at each of four household objects ("objects") of four different materials, with each distinct material represented in exactly one element of each category. We recorded each impact with a Shure SM57 microphone. Photos of the robotic setup and the tools and objects are shown in Figure 4. We filtered out any samples where the recording was too loud and clipped, which would obfuscate the frequency content, then used the two loudest remaining recordings of each pairwise interaction for our experiments, to ensure a high signal to noise ratio.

**Optimization setup**   We used the yellow elements of the model shown in Figure 1. Compared to the description in Section 3, we modified the model as follows: 1) instead of using the neural encoder to control impact force, the impact force profile for each instance was a single impulse parameterized by the Impulse trainable parameters, and 2) the Tool parameters were also used to produce a modal impulse response which was convolved with the force profile along with the Object modal impulse response. The parameters for each tool and each object were optimized with respect to each of the recordings for which they were present. All recordings shared the same set of Noise parameters, with each having a single gain scalar for the noise. We then performed analysis by synthesis on the entire batch of 32 recordings at once, optimizing all of our trainable variables with the Adam optimizer with respect to the loss that only compares ground truth with generated spectrograms of the non-separated

sound. For producing the separated object sound, we turned off the background noise, reverberation, acceleration sound, and the tool impulse response, then generated the model output for each recording. For producing the separated tool sound, we turned off the background noise and the object impulse response, then generated the model output for each recording. See Appendix D for more details.

**Baselines** We compared with non-negative matrix factorization (NMF) clustering [38] and the more recent "Deep Audio Prior" (DAP) [39] networks. Similar to our framework, both baselines require no pre-training. Note that these baselines do not have a built-in way of specifying which separated source corresponds to the tool and which to the object, whereas ours does.

**Metrics** We provide qualitative examples of the separated sources in our Supplementary Materials. Ground truth separated object sounds are not available to compare our results against, as there is no realistic way to even collect them in the real world. Thus, for a quantitative comparison, we compare the performance of material classification on each framework's outputs. Specifically, we train an audio-based material classifier on balanced data of only the relevant materials from the Sound-20K dataset, which consists of only *synthetic* sounds of objects of different materials bouncing on surfaces in a virtual environment [40]. We tested this trained classifier on the original sound as well as the source-separated output of each framework, to test whether the separated audio can be more accurately classified by this trained material classifier.

**Results** We report results in Table 1. Because our baselines have no constraint on which separated source is the tool and which is the object, we show the results of two evaluation schemes. Under the "Dataset Max" scheme, we compute the object material accuracy twice for each baseline: first treating their first output as the object, then treating the second as the object. We use the higher between the two as their result.

**Table 1:** Classifying materials from source-separated audio.

| Model | Object Material Acc. (%) | | Tool Material Acc. (%) | |
|---|---|---|---|---|
| | Dataset Max | Inst. Max | Dataset Max | Inst. Max |
| Raw audio | 37.5 | 37.5 | 12.5 | 12.5 |
| NMF [38] | 40.6 | 43.8 | 21.9 | 21.9 |
| DAP [39] | 50.0 | 68.8 | 28.1 | 50.0 |
| DiffImpact (ours) | **90.6** | **96.9** | **43.8** | **53.1** |

We do the same for tools. Under the "Inst. Max" scheme, we take the maximum for each instance instead of across the whole dataset. While both schemes offer the baselines an advantage, our DiffImpact still outperformed them, while also explicitly differentiating between the object and the tool. All models, including ours, performed better for classifying objects than tools. Object sounds dominated the recordings over tool sounds, since they were much larger, with more surface area for dispersing vibrations. Their size also gave them stronger low frequency modes with lower dampings, lasting longer in recordings, and thus easier to capture and characterize. See Appendix D for confusion matrices from this task, as well as results from a similar experiment where our DiffImpact outperforms the same baselines for estimating Young's modulus of each object and tool from separated sounds.

## 5 Conclusion

We have proposed DiffImpact, a fully differentiable physics-based framework for modeling rigid object impact sounds. This model enables robots to learn valuable physically interpretable parameters from sounds of objects in their everyday environments. We first validated that our method successfully simulates impact sounds of different everyday objects, and that it can be used in an analysis-by-synthesis approach to estimate impact forces that generated the analysed sound. We also showed how our framework can be used as a neural rendering layer, learning physically interpretable parameters of object sound models during an end-to-end learning task from a real-world dataset. We evaluated its performance against state of the art audio learning baselines with both a human study and distance metrics, and found our model to be more expressive and resource efficient than alternatives for modeling real-world impact sounds. Finally, we showed how a robot can use our model to learn sound models of everyday objects in its environment by striking them in a noisy environment with everyday tools. The robot used our model to differentiate between the sounds of the tools it used and the objects it struck with them and could classify the materials of the objects it strikes from sound 140% more accurately than from the original, unseparated audio or from generic audio source separation baselines. The knowledge robots can use our model to ascertain about objects, materials, and contact forces will be valuable to future applications which use our approach in robot decision making and policy learning. As future work, we aim to expand the scope of our model to other contact events, such as rubbing and scratching, or other object acoustic events, such as bouncing on different surfaces or fracturing. We also intend to use our contributions as a foundation for models and applications which use other multimodal inputs, such as vision and tactile data.

**Acknowledgments**

We thank Christopher Atkeson, Ante Qu, Oliver Kroemer, Jacky Liang, and Leonid Keselman for valuable discussions, Kevin Zhang, Michael A. Lin, Hojung Choi, Brian Roberts, and Toki Migimatsu for assistance in setting up robotic experiments, and Matt Wright for assisting with audio recordings. This work is in part supported by the Toyota Research Institute (TRI), NSF CCRI #2120095, Amazon Research Award (ARA), Autodesk, IBM, and the Stanford Institute for Human-Centered AI (HAI). N. Heravi was supported by the NSF Graduate Research Fellowship.

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
