# OpenReview forum: "DiffImpact: Differentiable Rendering and Identification of Impact Sounds"
_robot-learning.org/CoRL/2021/Conference — CoRL2021 Oral_

### Official Review · Reviewer_FKcr · 2021-07-18

**Originality:** Very Good
**Technical Quality:** Excellent
**Clarity Of Presentation:** Excellent
**Impact:** 4

**Recommendation:**

Strong Accept: I recommend accepting the paper and will argue for my recommendation even if other reviewers hold a different opinion.

**Summary:**

The paper presents a network architecture (DiffImpact) and analytic modeling scheme for differentiable rendering of impact sounds for rigid objects. Analytic modeling choices such as impulse force and impulse response are taken from prior literature and adapted accordingly, such as taking a Gaussian approximation to impulse force and allowing for multiple contacts. Environment noise and reverb from the environment are also accounted for. The model is trained with ground truth spectrogram as input and there are separate generator modules for impact force profile, impulse response, environment noise, and environment reverb. The outputs of these modules are combined to produce a prediction of a final waveform, where an L1 loss supervises the learning against the ground truth spectrogram. Downstream tasks include reproducing waveforms from ASMR videos, sound separation for a robot hitting an object with a tool (i.e. generating separate waveforms for the impacted object and the tool the robot hit object with), and classifying object material. Both quantitative and qualitative results are shown, where the latter used a Mechanical Turk user study to compare preferences of the generated waveforms of DiffImpact to other baselines including a state-of-art speech generator. Qualitative results show users preferred DiffImpact over the baselines for all objects except a deformable object, which violates the rigid body assumptions of the paper. On the object material prediction task, DiffImpact (separating tool and object impact sounds) significantly improves classification accuracy over using the original waveforms.


**Issues:**

The use of a robot is limited, primarily just for automated data collection. It’s true material classification is a downstream task that is useful in robotics, but it would be more compelling to show a robot can perform a task in an improved way using the proposed method. I think it is sufficient that the proposed approach is promising for further research that can incorporate the approach into policy learning schemes, but perhaps more can be said in the paper about potential applications in autonomous robot learning.

**Reviewer Expertise:**

Good: General knowledge of the area

**Strengths And Weaknesses:**

Strengths:
+ The experimental results are very impressive, particularly separating the waveforms of the tool and object being impacted. This can be extremely useful for robot manipulation when the robot has prior knowledge about the tool (e.g. it’s just tapping with its own finger/end-effector).
+ The generated waveforms for the ASMR videos sound very realistic, effectively noise-filtered versions of the ground truth sounds for several of the objects. It’s also intuitive that the method fails on the deformable object given the method’s rigid body assumption.
+ The downstream task of material classification is a nice showcase of the method.
+ The paper is clearly written. I am not an expert in audio modeling, yet I could follow the approach very well and understand the technical details provided in the appendix. The paper provides sufficient high-level information in the main text and sufficient technical details in the appendix.
Weaknesses:
- It would be beneficial to show robot experiments that demonstrate the approach can benefit robot policy learning and decision-making in some way, as opposed to only using the robot to collect data. For example, a task where success is dependent on using an object that is composed of a particular material (e.g. a metallic object that is heavier than the others), and show DiffImpact enables finding and using that object better than other baselines.
- The model input is a spectrogram which depending on the task for the robot may have limited utility. An architecture that takes in multimodal inputs (e.g. the video, audio, force sensor readings, joint state, etc.) would provide more downstream flexibility and allow for cross-modal predictions.


**Summary Of Recommendation:**

While the robot experiments do not directly show the benefits of the approach for performing robot tasks, the prospective benefits of utilizing the method for downstream robot manipulation tasks is promising. Audio is a very under-utilized modality for real-robot manipulation, in part because of the difficulty in acquiring a useful signal that isn’t polluted by background noise and distractor sounds. This approach could feasibly allow the robot to reason in a targeted way about the audio signals generated only from its intentional interactions with object and filter out the unwanted distractors. Utilizing this method with other robot sensory modalities and connecting the features with the robot’s action space seems like a truly promising extension to this work, and would be tractable to pursue given it is a neural network that can be integrated to other architectures. I therefore think this approach has promise to inspire further research and extensions that could have a big impact on robot manipulation research.

---

> ### Author Response · Authors · 2021-08-29
> **Author Response to Reviewer FKcr**
>
> Thank you for your suggestions for how our contributions can be strengthened and extended!
>
> ### Q1: Discuss how this approach could be helpful for robot policy learning or decision making.
> We fully agree and hope that our work’s contributions to robotic perception of impact sounds will empower better reasoning and decision-making capabilities of robots in semi- and unstructured environments. As you’ve pointed out, we show that our framework outperforms baselines for classifying materials of objects and tools, and now due to our newly added experiment based on Reviewer 7DAN’s suggestions, we also show that our approach outperforms baselines in directly estimating Young’s modulus (stiffness) of object and tool materials. These capabilities can be applied to future work involving robot grasping, tool usage and affordance reasoning, and other tasks in learning for manipulation.
> **Revision**: We have added a comment to our Conclusion about the potential for future work in applying our approach to robot policy learning and decision-making.
>
> ### Q2: An architecture with multimodal inputs could be more flexible and powerful.
> We whole-heartedly agree here as well. From a practical engineering standpoint, we have structured our differentiable generators such that they can be used with different upstream inputs for future work, such as swapping out the spectrogram-based encoder network we used and instead using an encoder based on visual input (similar to “Visually Indicated Sounds” by Owens et al.) or based on the observable physical parameters of a system (similar to“STReSSD: Sim-To-Real from Sound for Stochastic Dynamics” by Matl et al.).
> **Revision**: We have added a comment to our Conclusion about the potential for future work in using multimodal inputs to our model for new tasks and applications.

---

### Official Review · Reviewer_8sMD · 2021-07-27

**Originality:** Excellent
**Technical Quality:** Excellent
**Clarity Of Presentation:** Excellent
**Impact:** 4

**Recommendation:**

Strong Accept: I recommend accepting the paper and will argue for my recommendation even if other reviewers hold a different opinion.

**Summary:**

The paper introduces a new, differentiable audio model that can generate the sound of one or more rigid objects being struck. The authors show that by differentiating through the acoustic model, they can estimate material properties that allow them to identify objects.

The math for the audio signal generation is slightly out of my league but otherwise, I found the paper great. With the rise of differentiable physics engines, this was a matter of time.
I couldn't find any major weakness that would prevent me from recommending the work.

**Issues:**

No issues beyond the numbered weaknesses listed above.

**Reviewer Expertise:**

Fair: Some knowledge of the area

**Strengths And Weaknesses:**

### Strengths

- Well-written, clear and detailed.
- Experiments are great, both the in-the-wild ASMR data, human preference ranking, and the Franka Panda hitting the objects. Lots of effort and very illustrative of the strengths of your model.
- Selection of baselines is good as far as I can tell. I'm no expert in audio generation but I liked the contrast of model-based and model-free methods.
- Appendix is thorough with experimental details.

### Weaknesses

1. I can't really judge the quality of the audio generator formulas. Maybe some better reviewer can elucidate if this makes sense. However, I would have appreciated code or at least a jupyter notebook showing a little demo of the generator in the supplementary material.
2. I've worked on a differentiable physics engine before and there it was the case that the system was very sensitive to the random initialization. You seemed to glance over that topic in a single sentence (line 205), without specifying the initialization ranges. Could you discuss a bit, please, if you found any issues here, e.g. if the model is diverging if a parameter is exceeding a certain distance from the GT, or if there are local minima that the model can get stuck in?
3. I'd appreciate if you could write the optimization strategy in the robotic setup (sec 4.3-Optimization Setup) into a pseudocode block for clarity. That part wasn't 100% clear to me.

**Summary Of Recommendation:**

I enjoyed the paper and couldn't find any major weaknesses. But I remain open to others having issues with the math or lack of reproducibility.

---

> ### Author Response · Authors · 2021-08-29
> **Author Response to Reviewer 8sMD**
>
> Thank you for posing some unique practical questions!
> ### Q1: Provide source code or a Jupyter notebook demonstrating audio generators.
> **Revision**: We have added commented Jupyter notebooks for both the analysis by synthesis (4.1) and source separation tasks (4.3) to our supplementary materials. We will release and document all our source code and data if our paper is accepted.
>
> ### Q2: Explain the model's sensitivity to initialization.
> Our model is indeed sensitive to careful initialization and a good choice of hyperparameters. For this reason, we structured our generators with hyperparameter biases which we tuned to balance each generator’s contribution to the impact sound at a coarse level. We also selected hyperparameters for the multi-scale spectrogram loss function to mitigate the potential for local minima on both the temporal and frequency dimension of the optimization landscape for aligning the synthetic impact sound with the ground truth recording.
> **Revision**: We have added a new section to the appendix (“Appendix F Hyperparameters and Initialization”).
>
> ### Q3: Add a pseudocode block to make the source separation algorithm (4.3) more clear.
> **Revision**: We have added pseudocode to the appendix (under “Appendix D Source Separation Experiment Details”), and we have also included the Jupyter notebook for this, for those interested in even more details.

---

### Official Review · Reviewer_7DAN · 2021-08-04

**Originality:** Excellent
**Technical Quality:** Very Good
**Clarity Of Presentation:** Excellent
**Impact:** 4

**Recommendation:**

Strong Accept: I recommend accepting the paper and will argue for my recommendation even if other reviewers hold a different opinion.

**Summary:**

This paper proposes a realization of the physical processes that generate "impact sound" (i.e., the sound produced by an object striking another) into a differentiable programming framework. Doing so enables computing accurate, and---importantly---interpretable solutions to inverse problems, such as estimating physical properties (contact forces), and of learning models of impact sounds from real-world audio. Results are also presented on a real robot, on a sound source seperation task.

**Issues:**

I have a few minor, yet relevant, issues (W1 through W3) that I would like to discuss in the revision phase.

**Reviewer Expertise:**

Very good: Comprehensive knowledge of the area

**Strengths And Weaknesses:**

Strengths
=========

**S1** Significant advance: This paper presents a significant advance in perception of impact sounds; and weaves together ideas from multiple disparate communities (ML, graphics, audio processing, robotics). Synthesizing realistic impact sounds is a yet-unsolved problem in the computer graphics and audio processing communities. Inverse problems like estimating physical parameters of contact from impact audio signals are currently tackled solely by "model-free" learning; these solution schemes lack physical interpretability. These are often compounded by the inherent noise in real-world audio recordings, and the combinatorially large sets of possible material interactions. Differentiable impact sound synthesis (DiffImpact) addresses the aforementioned problems in a single framework, by leveraging differentiable programming and gradient descent.

**S2** Novelty / Topicality / Relevance: This paper adopts an emerging trend of baking in inductive biases about physical processes into a parameterized learning modules; to improve physical interpretability and dramatically reduce sample complexity. Differentiable programming has seen success with inverse problems in vision, graphics, and optimization (differentiable rendering, simulation, and optimization). To the best of my knowledge, this paper is the first to apply these ideas to impact sound synthesis.

**S3** Breadth of experiments: I liked the generality of the framework; particularly the demonstration on real-world videos. The audio samples produced by DiffImpact (from the supplementary material) are of superior quality to other model-free baselines, demonstrating the need for inductive bias (differentiable impact sound modeling). As opposed to a pure system-identification approach (which is the common mode of deploying differentiable physics priors), DiffImpact trains an autoencoder for its impact sound modeling (Sec. 4.2), improving the quality of output audio.

**S4** Clarity / exposition: Overall, this paper was a joy to read -- very well-written and adequately positioned w.r.t. relevant prior work.


Weaknesses
==========

I have a few questions that I hope to see discussed during the review cycle.

**W1** Justification of assumed quantities: From the paper and the supplementary materials, it is unclear as to what quantities (and if applicable, what priors over these quantities) are assumed given. For the experiments in 4.1, is the microphone assumed to be static (and is a single microphone assumed)? More importantly, are all quantities fully observable? I would presume some pairs of quantities may not be co-observable (and therefore may not be jointly estimated; E.g., the material properties and contact force parameters). It would help to add a note / table to this effect.

An addendum to the above discussion point: is it technically feasible to retrieve physically grounded contact parameters (not just proportional quantities) by assuming some of the quantities are known (or a strong prior over their absolute values is provided?


**W2** Clarification on environmental audio: From the description of the room impulse response (Sec. 3.4), it seems like what's considered environment noise is actually reverberations of the impact sound *without regards to the geometry of the environment*? That is, the model does not account for the precise manner in which sound reflects off of various surfaces in the environment, but rather lumps all of these terms into a single "ambient noise"? If so, this could be made more explicit in the paper.


**W3** Granularity for the classification task: Arguably, among all tasks presented in this paper, the source separation task (Sec. 4.3) is one that has seen the most interest from the ML community. Classification accuracies for source separation rely on the set of object materials chosen. The four chosen materials are quite distinctive (and perhaps a small sample size). It would therefore make for a more interesting experiment to compare the absolute values of estimated material properties (as opposed to computing a classification accuracy).

**Summary Of Recommendation:**

In summary, this paper presents a novel idea that constitutes a significant advance in processing impact audio from microphones. This paper has the potential to spur several interesting follow-up directions. This paper is quite interdisciplinary as well, which I see as a promising sign.

---

> ### Author Response · Authors · 2021-08-29
> **Author Response to Reviewer 7DAN**
>
> Thank you for pointing out where we should be more clear, and especially for the new experiment suggestion!
>
> ### Q1: Justify assumed quantities and clarify observability.
>
> Many quantities have priors induced on them by the hyperparameters we select for biases, scaling function bounds, and initial values. The single mono-channel microphone is indeed assumed to be static. Not all quantities are fully observable in our dataset (e.g. microphone gain and distance), and you are correct that some quantities cannot therefore be jointly estimated to an absolute quantity (e.g. absolute impact magnitude and absolute object impulse response magnitudes). Because of these assumptions, our model can only estimate proportional magnitudes for physical contact parameters, though the time scales of these contact parameters (relevant to sharpness) can theoretically be estimated at an absolute scale.
> **Revision:** We have added a new section to the appendix ("Appendix E Model Assumptions") which expands on this brief summary, and we added a table as suggested.
>
> ### Q2: Clarify environmental audio (noise and reverb).
> The environment noise is a time-constant static noise. This is distinct from the reverberation sounds, an impulse response filter composed of exponentially decaying filtered noise which is convolved with the unreverberated impact sound to produce the reverberated sound. The resulting noise from reverberations is dependent on the impact sounds, whereas the environment noise is independent. You are correct that the environment impulse response is estimated without knowledge of the geometry of the environment. We do not know the distances, angles, and absorptions of nearby surfaces, and generally for recordings in the wild, we cannot measure the environment impulse response separately with a balloon pop test. We instead estimate the gains and decays of the late stage reverberation noises from our recordings. Though the ambient noise and the reverberation sounds both physically come from the environment rather than the object, they are not lumped together in our model's construction.
> **Revision:** We have more clearly distinguished between the ambient noise and the environment reverberation sounds in our "Methods" section, and added more details in the appendix on model assumptions.
>
> ### Q3: Estimating an absolute material property from the impact sound would be interesting.
> Thank you very much for this suggestion. We just performed such an experiment, where instead of using our separated impact sounds for material classification, we have used them to estimate Young's modulus in a regression task. Note that Young's modulus could be very difficult for a robot to estimate through other sensory modalities (vision, proprioception, touch, etc.), yet estimating how stiff an object or tool is may be important for judging its affordances.
> **Revision:** We have added a new sub-experiment to our source separation results of Section 4.3, using our model's separated audio as input to a regression model to predict Young's modulus. We give more details in the paper, including the following table of results:
>
> ||Object Dataset Min Error (%)|Object Instance Min Error (%)|Tool Dataset Min Error (%)|Tool Instance Min Error (%)|
> |--|:--:|:--:|:--:|:--:|
> | Raw audio |263.4 | 263.4 | 243.3 | 243.3|
> |NMF (Spiertz et al.)|174.4 | 93.4 | 153.3 | 97.8|
> |DAP (Tian et al.)|161.0 | 71.9 | 227.7 | 104.1|
> |DiffImpact (ours)|**59.6**|**46.1**|**118.0**|**86.6**|

---

### Meta-Review · Area_Chair_663o · 2021-08-13

**Recommendation:** Accept (Oral)
**Confidence:** 5

**Metareview:**

All three reviewers are very positive about this paper. Whilst the paper does not explicitly involve robotics experiments, the reviewers agree that there is good future potential in this direction, and furthermore, this is an interesting area of robotics which is rarely addressed by the community. The main weaknesses come from queries and requests for clarifications; please address these in the rebuttal to maintain these positive reviews.

-------

Following the reviews, the authors have addressed a number of queries by the reviewers, and all the reviewers have maintained their recommendation of acceptance. This paper presents an interesting solution for sound synthesis, with some encouraging results on material classification. Whilst the use of this method for actually controlling a robot is not yet clear, it may lead to interesting discussions in the community about how we should be using sound information for robotics, which still remains a relatively unexplored field.

---

> ### Author Response · Authors · 2021-08-29
> **Author Response to Area Chair 663o**
>
> Thank you all for these reviews. We appreciated each reviewer's attention to detail and complementary perspectives.
>
> We agreed with each reviewer's points, and we have responded below. We also made additions to the submission (currently in blue), including clarifying some language and discussing future work in the main body of the paper, adding new sections to the appendix to explain important details, attaching demo code, and adding a new experiment with interesting results. If our submission is accepted, we will release and document all our source code and data.

---

### Decision · Program_Chairs · 2021-09-13

**Decision:**

Accept (Oral)

**Comment:**

All three reviewers are very positive about this paper. Whilst the paper does not explicitly involve robotics experiments, the reviewers agree that there is good future potential in this direction, and furthermore, this is an interesting area of robotics which is rarely addressed by the community. The main weaknesses come from queries and requests for clarifications; please address these in the rebuttal to maintain these positive reviews.

-------

Following the reviews, the authors have addressed a number of queries by the reviewers, and all the reviewers have maintained their recommendation of acceptance. This paper presents an interesting solution for sound synthesis, with some encouraging results on material classification. Whilst the use of this method for actually controlling a robot is not yet clear, it may lead to interesting discussions in the community about how we should be using sound information for robotics, which still remains a relatively unexplored field.